# Immunomodulatory and Anti-Inflammatory Effects of Fucoidan: A Review

**DOI:** 10.3390/polym12102338

**Published:** 2020-10-13

**Authors:** Elisaveta Apostolova, Paolina Lukova, Alexandra Baldzhieva, Plamen Katsarov, Mariana Nikolova, Ilia Iliev, Lyudmil Peychev, Bogdan Trica, Florin Oancea, Cédric Delattre, Vesela Kokova

**Affiliations:** 1Department of Pharmacology and Drug Toxicology, Faculty of Pharmacy, Medical University-Plovdiv, Vasil Aprilov Str. 15A, 4002 Plovdiv, Bulgaria; elisaveta.apostolova@mu-plovdiv.bg (E.A.); lyudmil.peychev@mu-plovdiv.bg (L.P.); vesela.kokova@mu-plovdiv.bg (V.K.); 2Department of Pharmacognosy and Pharmaceutical Chemistry, Faculty of Pharmacy, Medical University-Plovdiv, Vasil Aprilov Str. 15A, 4002 Plovdiv, Bulgaria; 3Department of Microbiology and Immunology, Faculty of Pharmacy, Medical University-Plovdiv, Vasil Aprilov Str. 15A, 4002 Plovdiv, Bulgaria; alexandra.baldzhieva@mu-plovdiv.bg; 4Research Institute at Medical University-Plovdiv, Vasil Aprilov Str. 15A, 4002 Plovdiv, Bulgaria; plamen.katsarov@mu-plovdiv.bg; 5Department of Pharmaceutical Sciences, Faculty of Pharmacy, Medical University-Plovdiv, Vasil Aprilov Str. 15A, 4002 Plovdiv, Bulgaria; 6Department of Biochemistry and Microbiology, Faculty of Biology, Plovdiv University Paisii Hilendarski, Tsar Asen Str. 24, 4000 Plovdiv, Bulgaria; mariann_77@abv.bg (M.N.); ilievini@abv.bg (I.I.); 7Department of Bioresources, National Institute for Research & Development in Chemistry and Petrochemistry-ICECHIM Bucharest, Splaiul Independenței 202, 060021 Bucharest, Romania; trica.bogdan@gmail.com (B.T.); florin.oancea@icechim.ro (F.O.); 8CNRS, SIGMA Clermont, Institut Pascal, Université Clermont Auvergne, F-63000 Clermont-Ferrand, France; cedric.delattre@uca.fr; 9Institut Universitaire de France (IUF), 1 rue Descartes, 75005 Paris, France

**Keywords:** fucoidan, inflammation, immunomodulation, macroalgae, cytokines, polysaccharide

## Abstract

Inflammation is the initial response of the immune system to potentially harmful stimuli (e.g., injury, stress, and infections). The process involves activation of macrophages and neutrophils, which produce mediators, such as nitric oxide (NO), prostaglandin E2 (PGE2), pro-inflammatory and anti-inflammatory cytokines. The pro-inflammatory cytokines interleukin-1β (IL-1β), interleukin 6 (IL-6), and tumor necrosis factor-α (TNF-α) are considered as biomarkers of inflammation. Even though it occurs as a physiological defense mechanism, its involvement in the pathogenesis of various diseases is reported. Rheumatoid arthritis, inflammatory bowel disease, Alzheimer’s disease, and cardiovascular diseases are only a part of the diseases, in which pathogenesis the chronic inflammation is involved. Fucoidans are complex polysaccharides from brown seaweeds and some marine invertebrates, composed mainly of l-fucose and sulfate ester groups and minor amounts of neutral monosaccharides and uronic acids. Algae-derived fucoidans are studied intensively during the last years regarding their multiple biological activities and possible therapeutic potential. However, the source, species, molecular weight, composition, and structure of the polysaccharides, as well as the route of administration of fucoidans, could be crucial for their effects. Fucoidan is reported to act on different stages of the inflammatory process: (i) blocking of lymphocyte adhesion and invasion, (ii) inhibition of multiple enzymes, and (iii) induction of apoptosis. In this review, we focused on the immunemodulating and anti-inflammatory effects of fucoidans derived from macroalgae and the models used for their evaluation. Additional insights on the molecular structure of the compound are included.

## 1. Introduction

The inflammation is the initial response of the immune system to potentially harmful stimuli (e.g., injury, stress, and infections). Even though it occurs as a physiological defense mechanism, its involvement in the pathogenesis of various diseases is reported. Rheumatoid arthritis, inflammatory bowel disease, Alzheimer’s disease, cardiovascular diseases are only a part of the conditions, in which pathogenesis the chronic inflammation is involved [1,2]. Various inflammatory stimuli (e.g., bacterial endotoxin lipopolysaccharides (LPS) and other foreign antigens) could induce migration of macrophages and neutrophils to the site of contact. The activation of these cells leads to increased production and release of inflammatory mediators as nitric oxide (NO), prostaglandin E2 (PGE2), tumor necrosis factor-α (TNF-α), and interleukin-1β (IL-1β). The increased levels of these substances promote prolonged inflammation [2,3].

Fucoidans are a group of sulfated polysaccharides with abundant presence in the cell walls of brown seaweeds and other marine species [4]. Generally, the chemical composition of fucoidans from macroalgae is very complex and in high variance depending on the algal source, geographic location, and extraction process. Nevertheless, their structural backbone is composed of repeating *α*-(1→3) linked l-fucopyranose residues or alternating *α*-(1→3) and *α*-(1→4) linked l-fucopyranoses. The fucosyl residues could be mono- or disubstituted with sulfate and/or acetate groups on C-2 and C-4 or rarely in position C-3 [5,6]. Furthermore, along with the fucosyl main chain, a wide range of other monosaccharides (mannose, galactose, arabinose, xylose, glucose, etc.), uronic acids, and proteins may also be part of the fucoidan structure [7].

Fucoidans, derived from algae have been studied intensively during the last years regarding their multiple biological activities and possible therapeutic potential. Recently, various pharmacological effects, including antitumor, immunomodulatory, antiviral, antimicrobial, antidiabetic, nephroprotective, antioxidant, anti-inflammatory, and anticoagulant, have been reported [2,8,9]. Their beneficial effects on inflammatory diseases (pancreatitis, colitis, osteoarthritis, skin inflammation, etc.), neurodegenerative diseases, immune dysfunction, and tumors have been an object of intensive research. Moreover, fucoidans have been successfully used in numerous health conditions as diabetes, hepatic steatosis (fatty liver), liver fibrosis, renal ischemia, disrupted blood coagulation status, stem cell therapies, gastric ulcers, gout, bacterial and viral infections, snake bites, and others (Figure 1) [4,5,9,10,11,12].

The wide range of biological activities is the basis of multiple possible applications including immune responses modulating compounds, antiviral, anti-inflammatory, and anticancer preparations [10]. However, the development of standardized fucoidan supplements is a complicated process due to the strong influence of the source, species, molecular weight, composition, and structure of the molecules, as well as the route of administration on the efficacy of the compound. Moreover, these activities are reported mostly based on in vitro experiments or in vivo evaluations on experimental animals. Fucoidans isolated from the same algal source could have the opposite effect when evaluated in different in vitro models or different animal models.

In this review, we focus on the anti-inflammatory and immunemodulating effects of fucoidans derived from macroalgae and the models used for its activity evaluation. Additional insights on the molecular structure of the compound and the process of inflammation are included.

## 2. Materials and Methods

The review article is based on the literature found in the databases of PubMed, Web of knowledge, and Science Direct in the period 2000–2020 year. Overall, 80 sources were selected for the review. The choice of the publications was made on the basis of the relevance of the publications to the topic, the research methodology, the research results, and the year of publication. The cited publications include systematic reviews, research articles, and meta-analysis.

## 3. An Overview of Human Immune System and Inflammation

Foreign antigens induce the macrophages synthesis and release pro-inflammatory cytokines such as TNF-α, IL-1β, and interleukin 6 (IL-6). These cytokines augment the chemotaxis of monocytes, granulocytes, lymphocytes, and mast cells to the site of injury and support antigen elimination and tissue restoration. Nevertheless, increased infiltration and activation of these cells elevate the risk of tissue damage due to excessive inflammation and its main symptoms pain and edema [13,14]. During chronic inflammation upregulation of pro-inflammatory mediators is observed. The main inflammatory mediators are inducible NO synthase (iNOS), cyclooxygenase-2 (COX-2), PGE2, and pro-inflammatory cytokines such as TNF-α, IL-1β, and IL-6. Modulation of the levels of these cytokines could be used to influence cellular responses and could be related to the anti-inflammatory activity of phytochemicals. Another mediator in the process of inflammation is NO. Low levels of NO have protective effects against inflammation. However, some pathological conditions are accompanied by increased production of NO and subsequent cytotoxicity and tissue damage. These processes are regulated via an intricate system of signaling pathways that regulate cellular responses [15,16].

The activation of T cells leads to synthesis of wide variety of lymphokines (e.g., interleukin-2, interferon-γ (IFN-γ), etc.). Some of them induce B cell growth and differentiation and activation in a wide range of white blood cells [17].

Excessive inflammation progressively destroys healthy tissue structure and causes organ destruction. This leads to increased demand for safe, orally administered substances with anti-inflammatory properties [11]. A possible candidate is fucoidan, whose antioxidative, anticoagulant, anticancer, anti-inflammatory, and antiviral activities are largely discussed. Its beneficial effects on inflammatory diseases, ischemia, immune dysfunction, and tumors are an object of intensive research. A possible explanation of its anti-inflammatory effect is the attenuation of the activation of the NF-κB signaling pathway [4,14]. Other authors discuss the role of the mitogen-activated protein kinase (MAPK) cascade in the biological effects of fucoidan. The MAPK family consists of serine/threonine protein kinases that are present in many mammalian cells. The results of various experiments show the role of the MAPK cascade in gene expression, cell proliferation and differentiation, neuronal survival, and apoptosis. Each of the main MAPK pathways is predominantly involved in specific processes. For example, the p38 MAPK pathway regulates the synthesis and release of pro-inflammatory mediators; activation of the ERK pathway is essential for cell proliferation, survival, and differentiation; and activation of JNK pathway regulates apoptosis [18].

## 4. Structure and Structure—Activity Relationship of Fucoidan

Fucoidans represent a class of polysaccharides composed mainly of l-fucose and sulfate ester groups [12]. They are derived from the cell walls of brown algae (*Phaeophyceae*), seagrasses (*Cymodoceaceae*), and some marine invertebrates such as sea urchin (*Echinoidea*) and sea cucumbers (*Holothuroidea*) [19,20]. Basically, the chemical content of seaweed fucoidans depends on the algae species, their habitat, the season of collecting, and the extraction method [4]. *Fucus vesiculosus* fucoidans are characterized with simple structure, consisting of *α*-(1→3) linked l-fucose residues or of alternating α-(1→3) and *α*-(1→4) linked l-fucoses that may be sulfate substituted on C-4. The main chain could be linear or branched in every 2 or 3 fucose units with side chains built up of single fucoses or fucooligosaccharides [5]. Fucoidans obtained from other *Fucus* species (*F. evanescens*, *F. distinchus*, *F. serratus*) have additional acetate groups and small amounts of xylose and galactose. Moreover, the sulfate groups can appear not only on C-4 but also in positions 2 and 3 or including sometimes 2,4-disulfated fucoidans (Figure 2) [12,21,22].

The chemical composition of fucoidans from other brown seaweed genus is relatively more complex containing other neutral monosaccharides as galactose (Gal), xylose (Xyl), mannose (Man), glucose (Glc), rhamnose (Rha), arabinose (Ara), and uronic acids (mainly glucuronic acid (GlcA)), or they have a highly branched structure [4]. The extraction methodology could also have an impact on the fucoidan structure. Fucoidans are generally obtained by multiple-step extraction procedure using dilute mineral acid, water, or enzymes. Some novels, more efficient, and cost-saving physical techniques as microwave- or ultrasound-assisted extraction are also applied [23]. *Adenocytis utricularis* fucoidan obtained by water extraction at room temperature was reported as a galactofucan consisting of fucose, galactose, and sulfate groups. During extraction at higher temperature (70 °C), the fucoidan from the same brown seaweed was defined as uronofucoidan. It was built up of fucose, uronic acids, and minor amounts of neutral monosaccharides and sulfate esters [12,24]. Using hot water extraction, a linear fucoidan from *Fucus vesiculosus* was reported, whereas by using acid extraction, the obtained fucoidan was analyzed as a branched polysaccharide with ramifications in every 2 or 3 fucose residues [12]. The difference in the chemical content of fucoidans could influence their biological activity, including their anti-inflammatory and immunomodulatory mechanisms of action. *Undaria pinnatifida* fucoidan, e.g., contains fucose and galactose in ratio 1.1:1.0 and minor amounts of uronic acids. This fucoidan is reported to increase the levels of IFN-γ, but not significant changes in the levels of IL-4, IL-6, TNF-α, and NF-κB were observed [25]. *Cladosiphon okamuranus* Tokida fucoidan is with higher content of uronic acid with ratio Fuc:GlcA 6:1. It leads to decreased levels of IL-6 and attenuated activation of the NF-κB signaling pathway [26]. *Ascophyllum nodosum* fucoidan is highly branched with Fuc:Xyl:Gal ratio relatively 10:1:1 and traces of glucose and mannose, and it is reported to increase the production of IL-6, IL-8, and TNF-α from neutrophils [27]. *Turbinaria decurrens* polysaccharide with ratio Fuc:Gal:Xyl:Man:Rha = 6.0:1.3:1.1:1.0:0.6 reduced the expression of genes of COX-2, IL-1β, and the NF-κB signaling pathway [28]. The presence of carbohydrate receptors on dendritic cells provides an explanation for the important role of fucoidan monosaccharide content on its biological activity. These receptors respond, e.g., to mannose and galactose [1].

The content of fucose and sulfate groups could also influence the anti-inflammatory and immunomodulatory activity of fucoidan. Menshova et al. [29] investigated the dependence between the activation of the complement system and the sulfate content of water-extractable fucoidans from *F. evanescens*, *L. japonica,* and *L. cichorioides.* They concluded that the sulfate content was not of a crucial importance for the complement activation, but they highlighted the positive influence of fucose residues in the polysaccharide structure for the activation of the alternative pathway of the complement [29]. Moreover, the sulfate content may influence the receptor binding and the following NO production in the anti-inflammatory response [1].

A control of optimum molecular weight (MW) values of fucoidan is also required to achieve a desired biological activity and therapeutic application [23,30]. Usually, fucoidans are high-molecular-weight polysaccharides with MW up to 1600 kDa. Usually, the large size of fucoidan molecules restricts their absorption and passage through cell membranes [23]. Low-molecular-weight polysaccharides obtained by different depolymerization techniques, during or after the process of extraction, showed some better pharmacological effects compared to high-molecular-weight ones. For example, in order to investigate the effects of high-molecular-weight fucoidan (HMWF) with MW = 100 ± 4 kDa and low-molecular-weight fucoidan (LMWF) with MW < 30 kDa on the pathogenesis of rheumatoid arthritis, Park et al. [31] tested their efficacy on a mouse collagen-induced arthritis model. The results indicated that the daily oral administration of HMWF enhanced the severity of arthritis, the inflammatory responses in the joint cartilage, and the levels of collagen-specific antibodies, whereas LMWF reduced arthritis through the suppression of Th1-mediated immune reactions [31]. Nevertheless, some very low-molecular-weight fractions (below 10 or 30 kDa) are reported as nonactive [23]. *Undaria pinnatifida* fractions with MW = 130 kDa increased the viability of spleen cells and the production of IFN-γ and NO, whereas fractions of 30 kDa with similar composition had low activity [23,32]. *Saccharina longicruris* galactofucan (MW = 638–1529 kDa) reduced fibroblast proliferation, but once depolymerized under 10 kDa had no effect on fibroblast cell growth and protein secretion [23,33]. Other authors reported similar anti-inflammatory effects for high- and low-molecular-weight fucoidans. For example, LMWF from *Sargassum hemiphyllum* with MW = 0.8 kDa and HMWF fraction from *Sargassum horneri* with MW > 30 kDa, both, in the same tested dose (100 µg/mL), decreased the levels of TNF-α and some interleukins [34,35]. Despite the molecular weight being one of the factors influencing the biological activity, a single universal relationship cannot be established and more studies are needed [23].

Even though the number of publications on algae fucoidans is constantly increasing, there are still limited studies illustrating the direct relationship between extraction methodology/structure/anti-inflammatory and immunomodulatory activity. In the current review, in Section 6 and Section 7 are presented the anti-inflammatory and immunomodulatory effects of fucoidans from different algae species, obtained by different extraction methodologies and with differences in their structure and chemical composition.

## 5. Pharmacokinetic of Fucoidan

Fucoidan exerts its activity not only in vitro but also in vivo. However, there are reports that indicate low plasma levels of this polysaccharide after oral intake [36,37]. On the other hand, there is also evidence for increased levels of fucoidan in particular organs. Nagamine et al. [38] reported higher levels of fucoidan derived from *Cladosiphon okamuranus* in the liver of rats fed with fucoidan for 2 weeks in comparison to the respective levels in blood serum of the animals. The authors explored also the mechanisms of fucoidan transport through the cell membrane. They suggested that active transport is the most probable way of transport [38,39].

The type of pharmaceutical formulation could also influence the pharmacokinetics of fucoidan. According to Kimura et al. [40], encapsulation of fucoidan in nanoparticles could elevate its cytotoxic activity and this result is related at least partially to increased permeability. Fucoidan (MW 750 kDa) isolated from *Fucus vesiculosus* has shown good skin-penetrating properties after topical application in rats. Moreover, topical application of a dose of 100 mg/kg bw resulted in prolonged half-life of the compound in comparison to intravenous delivery of the same dose [41].

Recently, Pozharitskaya et al. [42] performed a study on the pharmacokinetic of *Fucus vesiculosus* fucoidan after gastric delivery in rats. Thirty minutes after the application, the compound was detected in the plasma, the concentration reaching its maximum on the fourth hour. In humans, ingestion of 1 g resulted in detectable blood serum concentrations, which were lower than the levels reported in rats [37]. Here, the difference in the molecular weight of the studied fucoidan should be noted. Pozharitskaya et al. [42] studied fucoidan with MW of 713 kDa, whereas Tokita et al. [37] used fucoidan with lower MW (66 kDa). Another study also confirms the hypothesis of the key role of the MW for the absorption of fucoidan. Rapid absorption was registered after intravenous application of LMWF (7.6 kDa) in rabbits. However, it was also quickly eliminated (mainly during the first 2 h after the application) [43]. In contrast, HMWF remains in the blood circulation longer. Evaluating the tissue distribution of fucoidan, Pozharitskaya et al. [42] reported increased fucoidan levels in kidneys, spleen, and liver after oral administration of a single dose in rats. More recent research of these authors reveals linear pharmacokinetics of fucoidan after topical delivery of a dose in the range of 50–150 mg/kg. The study also indicates a possible deposition of fucoidan in the skin and striated muscle tissue during the initial stages (up to the 60th min) after the application. The authors hypothesize that the deposition of the compound in these tissues could be related to the observed prolonged half-life after topical delivery of fucoidan [41].

## 6. Anti-Inflammatory Activity of Fucoidan: Evidence from In Vitro and In Vivo Models

Fucoidan was reported to act on different stages of the inflammatory process: blocking of lymphocyte adhesion and invasion, inhibition of multiple enzymes, induction of apoptosis. The most discussed possible mechanism of action of fucoidan is the downregulation of MAPK and NF-κB signaling pathways and the following decrease in the production of pro-inflammatory cytokines (Figure 3). The role of the inhibition of selectins was also suggested by some authors [44,45].

### 6.1. In Vitro Studies of the Anti-Inflammatory Effect of Fucoidan on Cell Lines

#### 6.1.1. RAW 264.7 Macrophages

The anti-inflammatory activity of fucoidan is the subject of many studies. Most of them are focused on the in vitro activity of the compound with murine RAW 264.7 macrophages being the most used cell culture. Jeong et al. [2] reported decreased secretion of PGE2 in RAW 264.7 cells pretreated with LPS. Moreover, the polysaccharide impeded the nuclear accumulation of NF-κB p65 subunit and the degradation of IκBα. Fucoidan from *Fucus vesiculosus* also diminished the secretion of TNF-α and IL-1β in these cells and inhibited the neutrophil infiltration, which revealed its potential to suppress the early stages of the inflammation [2].

Another study focused on the effects of modified fucoidan on LPS-induced nitric oxide production by RAW264.7 macrophages. Methacrylated fucoidan diminished the NO release and CD86 expression. CD86 are costimulators of the interaction between antigen-presenting cells and T cells and increased level of CD86 leads to enhanced immune response. The effect of fucoidan on the elevated CD86 level after treatment with LPS and IFN-γ was similar to the activity of the anti-inflammatory cytokine IL-10 [46].

Fernando et al. [15] tested the effects of fucoidan fractions derived from *Chnoospora minima* on LPS-stimulated RAW 264.7 macrophages and found dose-dependent inhibition of the LPS-induced production of NO (IC50 27.82 µg/mL), iNOS, and COX-2 expression. Fucoidan also attenuated the increased synthesis of PGE2 and pro-inflammatory mediators such as TNF-α, IL-1β, and IL-6 [16]. Similar results are reported for fucoidan extracted from *Ecklonia cava*. The fractionated fucoidans significantly reduced the NO production and the levels of TNF-α, IL-1β, and IL-6 in LPS-stimulated RAW 264.7 cells [47].

Sanjeewa et al. [35] performed series of experiments of several compounds derived from the brown alga *Sargassum horneri*. HMWF attenuated the production of NO, PGE2, and the pro-inflammatory cytokines TNF-α and IL-6 in a murine macrophage cell line (RAW 264.7) treated with LPS. Anti-inflammatory activity was observed also for crude polysaccharides from the Celluclast enzyme digest. The compound decreased the production of NO, PGE2, TNF-α, and IL-1β in LPS-stimulated RAW 264.7 cell line. The expression of iNOS and COX2 was also attenuated in this model of in vitro inflammation. The authors also observed downregulation of NF-κB and MAPK signaling pathways [35]. Recently, the researchers reported similar results for a polysaccharide purified from the aforementioned brown algae. After LPS challenge, the isolated fraction reduced the pro-inflammatory cytokine synthesis in macrophages and downregulated the NF-κB and MAPK signaling pathways [48].

Recent research performed by Ni et al. [16] revealed insights on the mechanism of the anti-inflammatory activity of purified fucoidan from *Saccharina japonica*, also known as *Laminaria japonica*. The authors reported decreased NO production, iNOS, and COX-2 expression after fucoidan application on LPS-stimulated RAW 264.7 macrophage cells. A decrease was also observed in the levels of TNF-α, IL-1β, and IL-6 in comparison to nontreated LPS-stimulated RAW 264.7 cells. The study also showed that the anti-inflammatory activity of fucoidan is related to attenuation of the phosphorylation of MAPK and NF-κB signaling pathways [16]. Decreased TNF-α production in Caco-2/RAW 264.7 coculture cells was observed also by Mizuno et al. [49] in LPS-stimulated cells after treatment with *Laminaria japonica* fucoidan.

#### 6.1.2. Human Keratinocyte Cell Line (HaCaT)

Keratinocytes, which are abundant in the epidermis, produce pro-inflammatory cytokines and chemokines in response to pathogen invasion in the upper layers of the skin. The process leads to skin inflammation and possible subsequent chronification, resulting in atopic dermatitis. The pro-inflammatory cytokines IL-1β and IL-6 are produced by keratinocytes and play a key role in tissue infiltration with inflammatory cells, keratinocyte proliferation, and synthesis of other cytokines by keratinocytes. Treatment with fucoidan decreased the levels of IL-1β and IL-6 and suppressed the synthesis of inflammatory chemokines [50]. Recently, Lee et al. [51] reported attenuated activity of COX-2 and decreased production of PGE2, TNF-α, and IL-8 in the same in vitro model after treatment with *Laminaria japonica* extract.

#### 6.1.3. Rat Primary Microglia

Fucoidan isolated from *Laminaria japonica* evoked decreased synthesis of NO and expression of iNOS in LPS-activated primary microglia. The LPS-mediated morphological transformation of the microglia was also suppressed. The elevated level of iNOS triggered by inflammation increased the production of NO, and this process could be influenced by intracellular signaling molecules (e.g., tyrosine kinases, protein kinase C, and MAPKs) and transcription factors such as nuclear factor NF-κB and activator protein (AP)-1. The anti-inflammatory effect of fucoidan is probably mediated by the p38 pathway [18]. Another study focused on the change in the cytokine levels in BV2 microglial cells after treatment with *Fucus vesiculosus* fucoidan. The sulfated polysaccharide significantly decreased the levels of NO, PGE2, IL-1β, and TNF-α in LPS-activated microglial cells. The mechanism of the anti-inflammatory effect is related to inhibition of NF-κB, Akt, ERK, p38 MAPK, and JNK pathways [52].

#### 6.1.4. Caco-2 Cell Line and Caco-2/RAW 264.7 Coculture

Low-molecular-weight fucoidan (LMWF) isolated from *Sargassum hemiphyllum* reduced the LPS-induced inflammation in human intestinal epithelial cell cultures (Caco-2 cell line). The sulfated polysaccharide decreased the levels of the pro-inflammatory cytokines, TNF-α and IL-1β, and increased the levels of anti-inflammatory cytokines, IL-10 and IFN-γ. The authors suggest the role of the inhibition of the NF-kB pathway as a mechanism of reduction in TNF-α and IL-1β levels [34]. Fucoidan from *Laminaria japonica* and extract from *Fucus vesiculosus* decreased the levels of IL-8 in Caco-2/RAW 264.7 coculture and Caco-2 cells, respectively [49,53]. More recent research performed by Yang et al. [54] reveals attenuated production of NO in LPS-stimulated Caco-2 cells treated with *Laminaria japonica* extracts. The authors examined the effects of four extracts from the brown algae on the pro-inflammatory cytokines IL-6 and TNF-α. IL-6 was reduced by all four tested extracts, whereas TNF-α showed decreased level after treatment with two of them [54].

#### 6.1.5. Other Models

Fucoidan from *Cladosiphon okamuranus* Tokida reduced the elevated levels of IL-6 in CMT-93 cells after treatment with LPS and diminished the activation of the NF-κB signaling pathway [26].

Pozharitskaya et al. [55] reported inhibition of human recombinant cyclooxygenase COX-1 (IC50 27 µg/mL) and COX-2 (IC50 4.3 µg/mL), in vitro after treatment with *Fucus vesiculosus* fucoidan. Moreover, the compound was found more active towards COX2 isoform. Downregulation of MAPK p38 signaling pathway in LPS-activated human mononuclear U937 cells was also observed [55].

The main results of in vitro evaluations of the effects of fucoidan are summarized in Table 1.

### 6.2. In Vivo Studies of the Effects of Fucoidan on Inflammation and Cytokine Levels in Animal Models

#### 6.2.1. Diabetes Mellitus in Rodents

Aleissa et al. [9] registered elevated levels of pro-inflammatory cytokines (IL-1β, IL-6, and TNF-α) in a model of streptozotocin-induced diabetes mellitus in rats. Fucoidan isolated from *Saccharina japonica* diminished the toxic effects of diabetes and/or aflatoxin B1 on the liver and kidneys via reducing the blood glucose levels and serum levels of IL-1β, IL-6, and TNF-α [9]. Xu et al. [56] performed a similar research, however, they reported decreased kidney levels of IL-6 and no change in the TNF-α levels after treatment with LMWF from the same brown alga. A combination of LMW fucoidan from *Sargassum hemiphyllum* and fucoxanthin reduced the levels of TNF-α and IL-6 in diabetic mice [57].

#### 6.2.2. Chronic Colitis in Mice

Matsumoto et al. [26] tested the effect of fucoidans of different origins on a model of chronic colitis in mice. The authors reported decreased synthesis of IFN-γ and IL-6 and increased levels of IL-10 and transforming growth factor (TGF)-β in lamina propria of the colon after treatment with fucoidan derived from *Cladosiphon okamuranus* Tokida. Interestingly, the same changes were not observed in mice treated with *Fucus vesiculosus* fucoidan [26].

Another study evaluated the effect of fucoidan extracts of *Fucus vesiculosus* on dextran sulfate sodium-induced model of acute colitis. Oral intake of fucoidan extracts significantly lowered the levels of IL-1α, IL-1β, and IL-10 derived from the colon tissues in mice. Moreover, the infiltration of the colon tissues with inflammatory cells and the submucosal edema were also reduced [58].

#### 6.2.3. Inflammation in Zebrafish Embryos

Zebrafish animal model is an often-used model of human diseases. Fucoidan derived from *Chnoospora minima* decreased the production of NO, COX-2, and iNOS expression in LPS-stimulated zebrafish embryos [15]. Anti-inflammatory effect and reduced synthesis of NO after LPS challenge were also reported for fucoidan extracted from *Ecklonia cava* [59]. Similar results were obtained by Sanjeewa et al. [48] for the HMWF-purified polysaccharide, obtained from brown macroalga *Sargassum horneri,* and one of the polysaccharide fractions, isolated from the brown alga, which decreased the NO production in LPS-stimulated embryos.

#### 6.2.4. Rheumatoid Arthritis in Mice and Rats

Low-molecular-weight fucoidan fractions from *Undaria pinnatifida* reduced cartilage and bone destruction and tissue infiltration with inflammatory cells in a model of collagen-induced rheumatoid arthritis in mice. Interestingly, HMWF had the opposite effect [31]. In the same animal model of arthritis, *Sargassum muticum* fucoidan decreased the paw edema. Moreover, the brown alga extract reduced the elevated levels of TNF-α, IFN-γ, and IL-6 in the serum [60].

Fucoidan derived from *Undaria pinnatifida* reduced the inflammation in complete Freund’s adjuvant-induced arthritis in rats [61]. Ananthi et al. [62] evaluated the effect of *Turbinaria ornata* fucoidan in the same model. Blood plasma levels of TNF-α, IL-6, and PGE2 were elevated in the rats, treated only with complete Freund’s adjuvant. Their respective levels in fucoidan-treated animals were reduced, reaching similar levels to the normal ones of these cytokines [62].

#### 6.2.5. Murine Paw Edema

Manikandan et al. [28] reported well-defined anti-inflammatory effect of fucoidan derived from *Turbinaria decurrens* on formalin-induced paw edema in mice. The sulfated polysaccharide reduced the expression of genes of COX-2, IL-1β, and the NF-κB signaling pathway. Similar effect of fucoidan isolated from *Undaria pinnatifida* was reported by Phull and Kim [51] in carrageenan-induced paw edema. Fucoidan from *Cystoseira sedoides*, *C. compressa*, and *C. crinita* also reduced the inflammation in this model of paw inflammation [63]. Anti-inflammatory effect in carrageenan-induced paw edema after treatment with fucoidan (from *Turbinaria ornata*) was observed by Ananthi et al. [64].

#### 6.2.6. Myocardial Infarction in Rats

Li et al. [65] studied the effects of *Laminaria japonica* fucoidan on the cytokine levels in a rat model of myocardial infarction. Ligation of the left anterior descending (LAD) coronary artery for 30 min induces ischemia and is followed by reperfusion with duration of 2 h. In this condition, the ischemia-reperfusion injury is followed by infiltration of the affected zone with leukocytes, which produce pro-inflammatory cytokines. The local inflammation results in myocardial cell damage. Fucoidan treatment significantly decreased the blood serum levels of TNF-α and IL-6 in rats with a model of myocardial infarction. Moreover, the sulfated polysaccharide increased the levels of IL-10, which is known as an anti-inflammatory cytokine in heart diseases. The authors concluded that the protective effects of fucoidan were a result of its influence on the pro- and anti-inflammatory cytokines [65]. Recently, Lu et al. [66] reported similar anti-inflammatory activity of fucoidan in similar model of acute myocardial infarction in rats. The authors reported decreased activity of NF-κB signaling pathway and reduced levels of TNF-α and IL-6 in fucoidan-treated animals [66].

#### 6.2.7. Liver Damage in Rodents

Inflammation-mediated fibrosis could occur because of chronic intake of alcohol. Transforming growth factor-β1 (TGF-β1) is an inflammatory mediator and a cytokine, which possess fibrogenic activity. Fucoidan from *Fucus vesiculosus* reduced the ethanol-induced liver injury and attenuated the hepatic production of inflammatory cytokines, such as TGF-β1, COX-2, and NO in alcohol-induced liver damage in mice [67]. Meenakshi et al. [68] also reported hepatoprotective effects of fucoidan, isolated from *Turbinaria decurrens*. The authors evaluated the effect of the sulfated polysaccharide on alcohol-induced liver damage in rats, however, their research was focused on the antioxidant effect of the compound [68].

Recently, AlKahtane et al. [69] revealed the anti-inflammatory effect of fucoidan in microcystin-LR-induced liver damage in mice. Mice treated with fucoidan isolated from *Laminaria japonica* for 21 days (50 or 100 mg/kg/day) showed significantly reduced serum levels of TNF-α, IL-1β, and IL-6 in comparison to animals treated only with the hepatotoxic agent. Moreover, the levels of TNF-α and IL-1β in mice, which received the higher dose of fucoidan (100 mg/kg/day), were similar to the controls [69].

Chale-Dzul et al. [70] evaluated the effect of fucoidan, derived from *Sargassum fluitans* Borgesen, on a model of carbon tetrachloride-induced liver injury in rats. Treatment with the sulfated polysaccharide significantly lowered the levels of TNF-α and IL-1β compared to rats treated only with carbon tetrachloride [70].

#### 6.2.8. Other Models

Fucoidan from *Cladosiphon okamuranus* decreased the neutrophil infiltration of the peritoneal cavity in a model of acute peritonitis in rats evaluated the effect of fucoidan, derived from *Kjellmaniella crassifolia* on aspirin-induced gastric ulcers in rats [71,72]. They reported decreased levels of pro-inflammatory cytokines (TNF-α, IL-1β, and IL-6) and increased levels of the anti-inflammatory cytokine IL-10 in rats’ serum. They also stated that this effect is probably related to downregulation of the NF-κB signaling pathway [72].

Park et al. [73] used an animal model of periodontitis induced by injection of LPS and *Porphyromonas gingivalis* to evaluate the effect of fucoidan isolated from *Fucus vesiculosus* in mice. The treatment evoked an anti-inflammatory effect, but no effect on the expression of pro-inflammatory cytokines and bacterial clearance was detected. The only cytokine, which was downregulated by fucoidan, was IFN-γ. The infiltration of monocytes and dendritic cells into the gingiva was reduced in fucoidan-treated mice. However, no changes in the expression of COX-2 were registered [73].

Aqueous extract of brown macroalga *Turbinaria ornata* exerted antioxidative activity in the cotton pellet model of chronic inflammation in rats. The extract, which contained sulfated polysaccharide, reduced the oxidative stress and inflammatory biomarkers [74].

Fucoidan from *Fucus vesiculosus* was found effective in the treatment of acute pancreatitis in mice. Decreased levels of IL-1β, TNF-α, and myeloperoxidase and reduced severity of histological changes were observed in two animal models of pancreatitis after injection of fucoidan. The neutrophil migration was also attenuated [45].

The main results of in vivo evaluations of fucoidan effects are listed in Table 2.

### 6.3. Clinical Trials

Myers et al. [75] reported the results of a pilot open label randomized clinical trial in patients with osteoarthritis. The study was performed on 12 participants (5 females and 7 males) diagnosed with knee osteoarthritis. The mean age of the females was 62 years (± 11.06) and the mean age of the males was 57.14 (± 9.20). Five participants received dose of 100 mg and 7 participants received dose of 1000 mg Maritech^®^ once daily after meal. Maritech^®^ is a combination of *Fucus vesiculosus* (85% *w/w*), *Macrocystis pyrifera* (10% *w/w*), *Laminaria japonica* (5% *w/w*), vitamin B6, zinc, and manganese. Oral intake of the composition for 12 weeks (only one participant was treated for 10 weeks) reduced the pain, stiffness, difficulty with physical activity, and overall symptom severity. The participants had comprehensive osteoarthritis test (COAT) scores between 3 and 7. The average score dropped from 4.54 (3.03–6.06) to 3.72 (2.24–5.20) in the low-dose group and from 4.81 (3.52–6.09) to 2.32 (1.07–3.57) at the end of the study. Data were presented as mean and 95% confidence intervals. Significant reduction was observed in the physical difficulties and overall symptoms subscales with more prominent differences in the higher dose (1000 mg). The scores of pain and stiffness did not show significant changes. However, the plasma level of TNF-α as an inflammation biomarker was not influenced. Further research of the authors explored the effect solely of *F. vesiculosus* extract on this condition and no significant improvement was found in comparison to placebo [10].

## 7. Immunomodulatory Effect of Fucoidan

Even though the anti-inflammatory activity of fucoidan is intensively reported, there are also studies revealing enhanced production of pro-inflammatory cytokines. Purified fucoidan from the brown seaweed *Undaria pinnatifida* promoted activation of human neutrophils, natural killer cells (NK), and production of pro-inflammatory cytokines (IL-6, IL-8, and TNF-α) and delayed their spontaneous apoptosis. Fucoidan derived from *Fucus vesiculosus* enhanced the maturation of dendritic cells, the activation of cytotoxic T cells, the Th1 immune responses, antibody production after antigen challenge, and production of memory T cells [76]. *Laminaria japonica, Laminaria cichorioides*, and *Fucus evanescens* fucoidans also could activate the immune defense. Makarenkova et al. [77] demonstrated the interaction of fucoidans with “toll-like receptors” (TLRs), which led to increased production of cytokines, chemokines, and expression of MHC molecules. The result was enhanced activity of both specific and innate immune cells. Toll-like receptors are a part of the innate immune system and substances, which bind to TLR to activate the NF-κB signaling pathway. Fucoidans bind to TLR-2 and TLR-4, but not to TLR-5, and enhance the immune response [39,77]. Hayashi et al. [78] reported augmented T cell and natural killer (NK) cell activation, as well as increased production of pro-inflammatory cytokines, in mice infected with HSV-1 after treatment with fucoidan from *Undaria pinnatifida* (Table 3).

A comparative study on the effects of fucoidans obtained from *Ascophyllum nodosum*, *Macrocystis pyrifera*, *Undaria pinnatifida,* and *Fucus vesiculosus* revealed their immunomodulatory potential. All fucoidans significantly increased the production of IL-6, IL-8, and TNF-α from purified human neutrophils. They also delayed apoptosis, *M. pyrifera* and *U. pinnatifida* having the most prominent effects. Fucoidan from *M. pyrifera* was considered the most promising compound compared to three other fucoidans due to delayed neutrophil apoptosis and its potential to enhance mouse NK cell activation, dendritic cell maturation, T cell immune responses, antigen-specific antibody production and memory T cell generation [32].

Recently, Tabarsa et al. [79] evaluated the effect of *Nizamuddinia zanardinii* fucoidan on the RAW264.7 murine macrophage cell line and NK-92 cells. One of the fractions obtained from the brown seaweed showed well-defined immune-enhancing activity. This polysaccharide induced increased secretion of NO, TNF-α, IL-1β, and IL-6 and activation of NK cells, NF-κB, and MAPKs signaling pathways with subsequent release of TNF-α and INF-γ [79]. Fucoidan derived from *A. nodosum* and *F. vesiculosus* also induced NO synthesis and cytokine production via activation of NF-κB and AP-1 signaling pathways in RAW264.7 cells [80].

Exposure to ultraviolet B (UVB) light leads to local inflammation of the skin manifested with edema and dermal infiltration of leukocytes. Low-dose UVB irradiation decreased the production of cytokines such as TNF-α, IL-4, IL-6, and IFN-γ in vitro, which could be related to immunosuppression. Attenuated production of IFN-γ in the type 1 T-helper immunity was found also in vivo (Table 4). CD4^+^ T cells are also involved in the immunosuppressive activity of UVB. The exposure increased the production and release of TNF-α by keratinocytes and fibroblasts. Fucoidan from *Undaria pinnatifida* showed immune-modulatory properties and increased the lowered levels of IFN-γ. It also reduced the edema and the leukocyte migration to the skin. However, the authors reported no significant difference in the levels of IL-4, IL-6, and TNF-α and the expression of NF-κB between the experimental mice [25].

## 8. Conclusions

Acute and chronic inflammatory diseases, associated with bacterial or viral agents, physical traumas, immunological reactions, or metabolic syndromes, affect many people worldwide. The quest for natural remedies for anti-inflammatory treatment and prevention is gaining increasing popularity in contemporary medicine. In this review, the potential of fucoidan as a natural marine product that covers a wide range of immunomodulatory and anti-inflammatory effects was demonstrated. The in vitro anti-inflammatory activity of fucoidan from brown algae could rely on reduced synthesis of NO and expression of iNOS; decreased secretion of PGE2, TNF-α, IL-1β, IL-6, IL-8, CD86, and COX-2 expression; downregulation of MAPK, Akt, ERK, JNK, STAT1, and NF-κB signaling pathways; decreased phosphorylation of p38 and ERK MAPK; and increased levels of IL-10 and IFN-γ. In vivo investigations on rodents and zebrafish embryos showed that the anti-inflammatory effect of fucoidan could be due to reduced blood glucose level and serum levels of IL-1α, IL-1β, IL-6, IL-10, TNF-α, IFN-γ, PGE2, TGF-β1, and myeloperoxidase; decreased production of NO, ROS, and iNOS expression; reduced expression of genes of COX-2 and the NF-κB signaling pathway; inhibited neutrophil migration; and increased levels of IL-10. Depending on the brown algae species, the immunomodulatory activity of fucoidan could be related to specific activation of TLR-2 or TLR-4 and following activation of NF-κB pathways; increased production of IL-6, IL-8, IL-1β, NO, and TNF-α from neutrophils; and activation of NK cells, NF-κB, and MAPKs signaling pathways with subsequent release of TNF-α and INF-γ. Although the numerous published studies must be validated in humans, conclusively, fucoidan could be suggested as a possible natural remedy to treat, delay, or prevent conditions related to inflammation or immune system disorders.

## Figures and Tables

**Figure 1 polymers-12-02338-f001:**
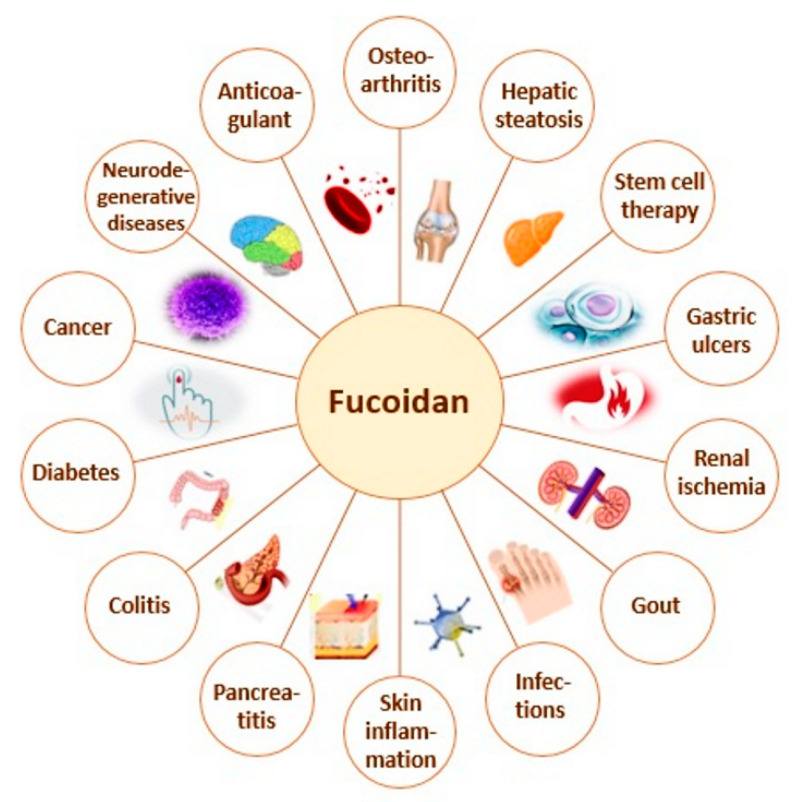
Proposed biomedical applications of fucoidan.

**Figure 2 polymers-12-02338-f002:**
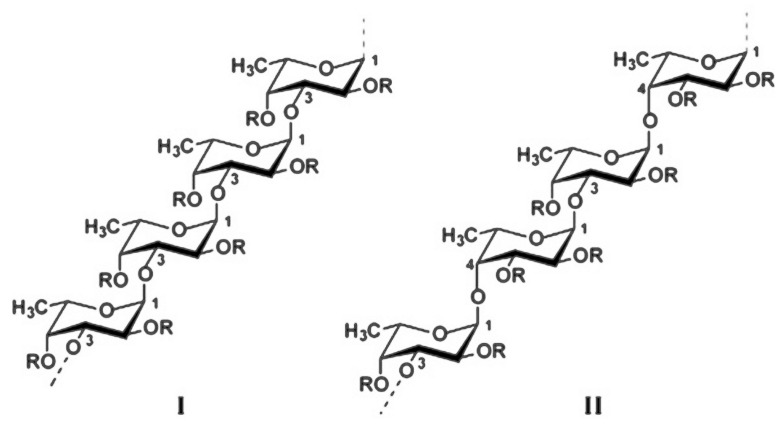
Fucoidans with *α*-(1→3) linked l-fucose residues (**I**) or alternating *α*-(1→3) and *α*-(1→4) linked l-fucoses (**II**). *R* represents the possible attachments of carbohydrate and noncarbohydrate substituents (sulfate and acetyl groups) [22].

**Figure 3 polymers-12-02338-f003:**
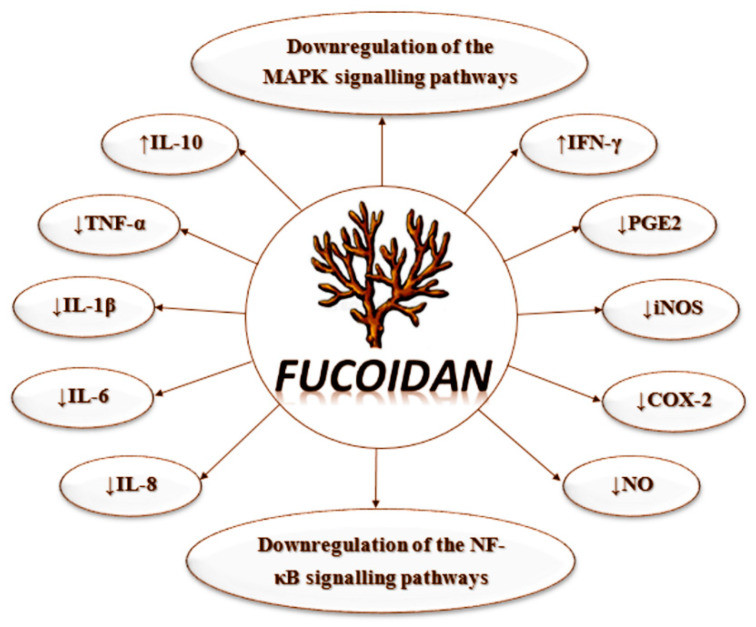
Putative mechanisms of anti-inflammatory activity of fucoidan.

**Table 1 polymers-12-02338-t001:** In vitro anti-inflammatory activity of fucoidan. Results were observed after LPS-stimulation of the cell culture.

Source	Compound	Cell Line	Concentration	Activity	Reference
*Fucus vesiculosus*	Commercially available fucoidan	Murine RAW 264.7 macrophages	50–100 µg/mL	Decreased secretion of NO, PGE2; diminished secretion of TNF-α and IL-1β	[2]
	Commercially available fucoidan	BV2 microglial cells	100 µg/mL	Decreased production of NO, PGE2, IL-1β, and TNF-α; inhibition of NF-κB, Akt, ERK, p38 MAPK, and JNK pathways	[52]
	Commercially available fucoidan	Human keratinocyte (HaCaT)	1–50 µg/mL	Inhibition of NF-κB, STAT1 and ERK ½ pathways; decreased production of IL-1β and IL-6 after TNF- α/ IFN-γ stimulation	[50]
	Methacrylate-functionalized fucoidan	RAW 264.7 macrophages; THP-1 macrophages	100 µg/mL	Decreased production of NO, decreased production of CD86 after LPS/IFN-γ stimulation	[46]
	Cold water extract	Caco-2	1 mg/mL	Decreased production of IL-8 after TNF-α stimulation	[53]
	Fuc:Glc:Gal:Xyl:Man = 1.0:0.16:0.05:0.09: 0.03:0.03	Human recombinant COX1 and COX2 assay kit	0.1–10 µg/mL	Inhibition of COX1 and COX2, predominantly of COX2	[55]
*Laminaria japonica*	LMWF (MW = 7 kDa); Fuc:Gal fraction = 1.0:0.24	Rat primary microglia	125 µg/mL	Reduced synthesis of NO and expression of iNOS; decreased phosphorylation of p38 and ERK MAPK	[18]
	HMWF containing Fuc 79.49%, Gal 16.76%, Rha 0.82%, Xyl 1.08%, Man 1.84%; sulfate content 30.72%; MW = 104.3 kDa	RAW 264.7 macrophages	25 µg/mL	Decreased production of TNF-α, IL-1β, IL-6, NO, iNOS, and COX-2 expression; downregulation of MAPK and NF-κB signaling pathways	[16]
	Lyophilized water extract	Human keratinocyte (HaCaT)	100 µg/mL	Decreased gene expression of COX2; diminished secretion of PGE2, TNF-α and IL-8 stimulated after UVB irradiation	[51]
	No information added	Caco-2/RAW 264.7 coculture	100 µg/mL	Decreased production of TNF-α and IL-8 mRNA expression	[49]
	Four types of water extracts (with or without fermentation)	Caco-2	100 µg/mL	Decreased production of NO and IL-6 by all four extracts	[54]
*Sargassum hemiphyllum*	LMWF (MW = 0.8 kDa), sulfate content 38.9%	Caco-2	100 µg/mL	Decreased TNF-α and IL-1β; increased IL-10 and IFN-γ	[34]
*Sargassum horneri*	HMWF; polysaccharide fraction (MW > 30 kDa)	RAW 264.7 macrophages	25–100 µg/mL	Decreased production of TNF-α, IL-6, NO and PGE2	[35]
*Chnoospora minima*	Fucose-rich polysaccharide fraction (79.32% Fuc)	RAW 264.7 macrophages	25–100 µg/mL	Decreased production of PGE2, TNF-α, IL-1β, IL-6, NO, iNOS and COX-2 expression	[15]
*Cladosiphon okamuranus* Tokida	Fucoidan containing Fuc:GlcA:sulfate groups = 6.1:1.0:2.9	Murine colon carcinoma cell line CMT-93	2.5 µg/mL	Decreased levels of IL-6; attenuated activation of the NF-κB signaling pathway	[26]
*Ecklonia cava*	LMWF rich in fucose (77.9% Fuc), sulfate content 39.1%	RAW 264.7 macrophages	50–100 µg/mL	Reduced NO production and levels of TNF-α, IL-1β, and IL-6	[47]

**Table 2 polymers-12-02338-t002:** In vivo anti-inflammatory activity of fucoidan.

Source	Composition	Model	DosesTested	Effective Doses	Positive Control	Treatment	Activity	Reference
*Fucus vesiculosus*	Extract 1: Fucoidan polyphenol complex (MW = 203 kDa, sulfate content 21.8%) Extract 2: high purity fucoidan (MW = 62 kDa, sulfate content 26.6%)	Acute colitis in C57BL/6 mice induced by oral intake of 3% *w/v* of dextran sulphate sodium for 7 consecutive days	Extract 1 in dose 400 mg/kg daily p.o.; Extract 2 in dose 10 mg/kg daily p.o. or i.p.	Extract 1 in dose 400 mg/kg daily p.o.; Extract 2 in dose 10 mg/kg daily p.o.	No	Oral or i.p. application for 7 days	Lowered levels of IL-1α, IL-1β, and IL-10	[58]
	Commercially available fucoidan	Acute pancreatitis in mice induced by cerulein or taurolithocholic acid sulfate	25 mg/kg	25 mg/kg	No	Intravenously; single dose, 30 min before application of cerulein or taurolithocholic acid sulfate	Decreased levels of IL-1β, TNF-α, and myeloperoxidase; inhibited neutrophil migration	[45]
	Commercially available fucoidan	Alcohol-induced liver damage in mice	30 and 60 mg/kg	30 and 60 mg/kg	No	Orally; 7 days of application	Decreased production of TGF-β1 and COX-2 in mice livers	[67]
*Laminaria japonica*	Commercially available fucoidan	Microcystin-LR-induced liver damage in mice	50 and 100 mg/kg	50 and 100 mg/kg	No	Orally; 21 days of application	Reduced levels of TNF-α, IL-1β, and IL-6	[69]
	No information	Myocardial infarction in rats	50, 100, and 200 mg/kg	100 and 200 mg/kg	No	Orally; 7 days of pretreatment before the surgery	Decreased levels of TNF- α and IL-6; increased levels of IL-10	[65]
	Commercially available fucoidan	Diabetes mellitus in rats, induced by 50 mg/kg STZ (i.p.)	100 mg/kg	100 mg/kg	No	Orally; daily application between fifth to eighth week after STZ injection	Reduced blood glucose level and serum levels of IL-1β, IL-6, and TNF-α	[9]
*Sargassum muticum*	Ethyl alcohol (70%) extract	Collagen-induced rheumatoid arthritis in mice	50, 100, and 200 mg/kg	100 and 200 mg/kg reduce TNF-α and IL-6 levels; 50, 100, and 200 mg/kg reduce TNF-α, IFN-γ, and IL-6	Joins^®^ 10 mg/kg	Orally; 77-day treatment (21–98 day of the experiment)	Decreased levels of TNF-α, IFN-γ, and IL-6	[60]
*Sargassum fluitans* Borgesen	Water extract, 7.56% sulfate content	Carbon tetrachloride-induced liver injury in rats	50 mg/kg	50 mg/kg	Silymarin 100 mg/kg p.o.	Orally; 1 week of pretreatment + 12 weeks treatment	Decreased levels of TNF-α and IL-1β	[70]
*Cladosiphon okamuranus* Tokida	Fucoidan containing Fuc:GlcA:sulfate groups = 6.1:1.0:2.9	Chronic colitis in mice induced by 4% dextran sodium sulphate (DSS) (p.o.)	No information	No information	No	Orally, as fucoidan containing chow; treatment during the whole 2-week experiment	Decreased levels of IL-6 and increased levels of IL-10	[26]
*Chnoospora minima*	Fucose-rich polysaccharide fraction (79.32% Fuc)	LPS-stimulated zebrafish embryos	12.5, 25, and 50 μg/mL added to the embryo media	12.5, 25, and 50 μg/mL	No	1-h long treatment on the 8th hour of postfertilization	Decreased production of NO, ROS, COX-2, and iNOS expression	[15]
*Ecklonia cava*	Extract (sulfate content 20.1%, Fuc 61.1%, Rha 3.9%, Gal 27.2%, Glc 0.8%, and Xyl 7%)	LPS-stimulated zebrafish embryos	100 μg/mL added to the embryo media	100 μg/mL added to the embryo media	Commercially available fucoidan	1-h long treatment on the 8th hour of postfertilization	Reduced production of NO	[59]
*Undaria pinnatifida*	LMWF, medium molecular weight fucoidan (MMWF), and HMWF	Collagen-induced rheumatoid arthritis in mice	300 mg/kg	300 mg/kg	No	Orally; 49 days treatment	LMWF reduced cartilage and bone destruction, and the tissue infiltration with inflammatory cells	[31]
*Turbinaria ornata*	Water extract and sulfated polysaccharide	Complete Freund’s adjuvant (CFA)-induced arthritis in rats	Water extract: 30, 100, and 300 mg/kg; sulfated polysaccharide 2.5, 5, and 10 mg/kg	Water extract 100 mg/kg and sulfated polysaccharide 5 and 10 mg/kg decreased TNF-α, IL-6, and PGE2 levels	Dexamethasone 100 μg/kg p.o.	Orally; pretreatment: two times a day for 7 days before CFA injection	Decreased levels of TNF-α, IL-6, and PGE2	[62]
*Turbinaria decurrens*	Extract; sulfate content 23.51%, Fuc 59.3%, Gal 12.6%, Man 9.6% Rha 6.4%, Xyl 11.4%	Formalin-induced paw edema in mice	50 mg/kg	50 mg/kg	Dexamethasone 2.5 mg/kg i.p.	Orally; 5 days of pretreatment	Reduced the expression of genes of COX-2, IL-1β, the NF-κB signaling pathway	[28]

**Table 3 polymers-12-02338-t003:** In vitro immunomodulatory activity of fucoidan.

Source	Compound	Cell Line/Lines	Tested Concentrations	Activity	Reference
*Laminaria japonica*	Fuc:Gal:Man:Xyl:Glc = 65:20:8:4:3 MW = 10–30 kDa	Human embryonic kidney cells (HEK293-null, HEK293-TLR2/CD14, HEK293-hTLR4/CD14-MD2, and HEK293-hTLR5)	1 mg/mL; 100 µg/mL; 10 µg/mL; 1 µg/mL; 100 ng/mL; 10 ng/mL of each extract	Specific activation of Toll-like receptors (TLR) 2 and following activation of NF-κB pathways is observed for *L. japonica* fucoidan (1 mg/mL), *L. cichorioides* fucoidan (100 μg/mL and 1 mg/mL), and *F. evanescens* fucoidan (10 μg/mL^−1^ mg/mL); activation of TLR-4 and following activation of NF-κB pathways is registered for *L. japonica* fucoidan (100 μg/mL and 1 mg/mL), *L. cichorioides* fucoidan (10 μg/mL^−1^ mg/mL), and *F. evanescens* fucoidan (1 μg/mL^−1^ mg/mL).	[77]
*Laminaria cichorioides*	Completely sulfated fucoidan MW = 40–80 kDa				
*Fucus evanescens*	Gal:Xyl:Man = 70:9:10.8 MW = 40–60 kDa				
*Ascophyllum nodosum*	Fuc:Xyl:Glc:Man:Gal = 39.80:3.68:0.88:0.72:3.37	Human neutrophils	50 μg/mL	All fucoidans significantly increased the production of IL-6, IL-8, and TNF-α from neutrophils	[27]
*Macrocystis pyrifera*	Fuc:Xyl:Glc:Man:Gal = 25.77:0.84:1.14:1.12:3.93				
*Undaria pinnatifida*	Fuc:Xyl:Glc:Man:Gal = 28.27:0.45:0.49:0.30:24.94				
*Fucus vesiculosus*	Fuc:Xyl:Glc:Man:Gal = 38.02:2.73:0.49:1.27:3.38				
*Nizamuddinia zanardinii*	Fuc:Xyl:Man:Gal = 38.1:15.2:33.2:13.4	RAW 264.7 murine macrophage	10, 25, and 50 μg/mL	Increased secretion of NO, TNF-α, IL-1β, and IL-6; activation of NK cells, NF-κB, and MAPKs signaling pathways with subsequent release of TNF-α and INF-γ	[79]

**Table 4 polymers-12-02338-t004:** In vivo immunomodulatory activity of fucoidan.

Source	Composition	Model	Dose Tested	Effective Dose	Positive Control	Treatment	Activity	Reference
*Fucus vesiculosus*	Commercially available fucoidan	C57BL/6 mice	10 mg/kg	10 mg/kg	No	Single dose injected i.p.	Increased levels of TNF-α and IL-6 in spleens and blood serum	[76]
*Undaria pinnatifida*	Fuc:Gal = 1.0:1.1; MW = 9000 Da	5-fluorouracil induces immunosuppression in BALB/c mice followed by herpes simplex virus-1 inoculation	10 mg/mouse per day	10 mg/mouse per day	No	7 days of oral treatment	Increased activity of NK cells	[78]
*Ascophyllum nodosum*	Fuc:Xyl:Glc:Man:Gal = 39.80:3.68: 0.88:0.72:3.37	C57BL/6 mice	50 mg/kg	50 mg/kg	No	4 days of intraperitoneal treatment	*M. pyrifera* fucoidan increased the maturation and activation of NK cells	[27]
*Macrocystis pyrifera*	Fuc:Xyl:Glc:Man:Gal = 25.77:0.84: 1.14:1.12:3.93							
*Undaria pinnatifida*	Fuc:Xyl:Glc:Man:Gal = 28.27:0.45: 0.49:0.30:24.94							
*Fucus vesiculosus*	Fuc:Xyl:Glc:Man:Gal = 38.02:2.73: 0.49:1.27:3.38							
*Undaria pinnatifida*	Gal 20.7%, Fuc 24.4%, uronic acid 8.2% and ester sulfate 31.3%	Skin of BALB/c mice after UVB irradiation	500 mg/kg	500 mg/kg	Vit. C 600 mg/kg weight	12 days of oral treatment	Increased lowered levels of IFN-γ after irradiation; reduced skin edema and leukocyte migration; no significant changes in IL-4, IL-6, TNF-α, and NF-κB expression	[25]

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
