# Peer review of "Immunomodulatory and Anti-Inflammatory Effects of Fucoidan: A Review"

_polymers, 2020, doi:10.3390/polym12102338_

Round 1

Reviewer 1 Report

Elisaveta Apostolova et al have submitted the review manuscript entitled "Immunomodulatory and anti-inflammatory effects of  fucoidan: a review". After close evaluation of the paper I would suggest revision according to the next points:

  1. According to the title the review should be focused in Immunomodulatory and anti-inflammatory effects of  fucoidan. Therefore there are a lot of common theoretical information about inflammation and immune system in Introduction. Please refocus it according to the title. The section 2.1 is not suitable. In Section 2.2. lines 125-150 are out of scope of paper.
  2. The Section 3 should be rewrited and should provide insight on relation between structure / extraction technology and future anti0inflammatory and immunomodulating properties. For example, authors provide statements "Macrocytis pyrifera fucoidan for example contains fucose and galactose in ratio 18:1 and traces of xylose. Ascophyllum nodosum fucoidan is highly branched with fucose:xylose:glucuronic acid ratio 5:1:1. Sargassum stenophyllum fucoidan has a wide range of components as fucose, galactose, mannose, glucuronic acid, glucose, 182 xylose, and sulfates [20]" However, this statement is not commented. How this related to anti-inflammatory and immunomodulating activity?
  3. The phrase "The large size of fucoidan molecules restricts their  absorption and passage through cell membranes which could influence their biological activity [25]" need clarification, how it "influence their biological activity"?
  4. Sections 4 and 5 are ;lacking of important details. Authors should discuss the relation of fucoidan structure (Mw, sulfate content, monosacharides, etc) on activity. The phrases like "Fucoidan from Fucus vesiculosus also diminished the secretion of TNF-α and IL-1β in these cells and inhibited the neutrophil infiltration..."; "Methacrylated fucoidan diminished the NO release and CD86 expression. CD86 are co-stimulators of the interaction between antigen-presenting cells and T  cells and increased level of CD86 leads to enhanced immune response..."; "Fucoidan also attenuated the increased synthesis of  PGE2 and pro-inflammatory mediators such as TNF-α, IL-1β, and IL-6..." etc are not informative. What was a structure of fucoidans mentioned, at which doses activity was observed, what was IC50??? Al these details should be provided. For improving of this part I suggest to authors to prepare a tables for each sections 4.1.1-4.1.5 in which indicate source of fucoidan; its composition; concentration tested, IC50 or concentration range in which activity was observed; type of activity (inhibition, stumulation, etc).
  5. For sections 4.2.1-4.2.8 I would suggest to summarise data in tables in which indicate fucoidan source, fucoudan composition, doses tested, activity observed in numbers. For example "Fucoidan treatment significantly decreased the blood serum levels of TNF-α and IL-6 in rats with a model of myocardial infarction" - how model of miocardial infarction was formed? which doses of fucoidan were tested? how fucoidan was administered to animals, was it single or multiple doses administration? for how many % "decreased the blood serum levels", are there positive control? what was efficacy of control comparing with fucoidan?. All these data should be included in the tables.
  6. The section 4.3 should be extended with data about type of clinical trial (randomised, placebo controlled, open/blind, number of patients, distribution (male/famale, age), doses, treatment schedule, etc. The resulkts should be provided in numbers as well. How many patients responded on therapy, "reduced the pain, stiffness, difficulty with physical activity and overall symptom severity..." - please provide real data in numbers.
  7. Similar data are required for section 5.
  8. The conclusion should be rewrited based on results presented in suggested Tables.
  9. The section Material  ad methods is lacking. How authors have collected information? Which sources were used, which time frame was used for literature.
  10. I would suggest to include in the review and discuss results described in the next recently published papers: https://doi.org/10.1016/j.ijbiomac.2020.04.012; https://doi.org/10.3389/fmars.2016.00129; https://doi.org/10.3390/md18050275; ;  https://doi.org/10.4162/nrp.2017.11.1.3 
  11. I would suggest to include special sub-section about pharmacokinetic of fucoidan. Pharmacokinetic help with interpretation of results observed.The bioavailability is essential for understanding of any pharmacological activity.

Author Response

Response to Reviewer 1:

The authors would like to thank the Reviewer 1 for the detailed description of all necessary changes in the manuscript. It was really helpful for us and all suggested corrections allowed us to improve the quality of this manuscript.

Please find below our answers to the comments of the Reviewer 1. All corrections, made in the revised manuscript, are marked in blue color.

Point 1. According to the title the review should be focused in Immunomodulatory and anti-inflammatory effects of fucoidan. Therefore there are a lot of common theoretical information about inflammation and immune system in Introduction. Please refocus it according to the title. The section 2.1 is not suitable. In Section 2.2. lines 125-150 are out of scope of paper.

Reply to point 1: We removed all details, which are not related to the mechanisms of anti-inflammatory and immunomodulatory activity of fucoidan from Introduction and Section 2. The structure of the whole Section 2 was changed. However, we would like to keep some sentences from lines 125-150, which provide basic information about pro- and anti-inflammatory cytokines. We believe that this information is necessary for the following explanation for the pharmacological effects of fucoidan.

Point 2. The Section 3 should be rewrited and should provide insight on relation between structure/extraction technology and future anti0inflammatory and immunomodulating properties. For example, authors provide statements "Macrocytis pyrifera fucoidan for example contains fucose and galactose in ratio 18:1 and traces of xylose. Ascophyllum nodosum fucoidan is highly branched with fucose:xylose:glucuronic acid ratio 5:1:1. Sargassum stenophyllum fucoidan has a wide range of components as fucose, galactose, mannose, glucuronic acid, glucose, 182 xylose, and sulfates [20]" However, this statement is not commented. How this related to anti-inflammatory and immunomodulating activity?

Reply to point 2: The Section 3 (in the revised manuscript section 4) is rewritten. Additional information about the anti-inflammatory and immunomodulatory mechanisms of fucoidan depending on its chemical content and Mw was added. Although there is limited information in the literature about the direct relationship between extraction methodology/chemical content/anti-inflammatory and immunomodulatory effects, we have tried to present the investigations of different scientific teams regarding the problem.

Point 3. The phrase "The large size of fucoidan molecules restricts their absorption and passage through cell membranes which could influence their biological activity [25]" need clarification, how it "influence their biological activity"?

Reply to point 3: Additional information was added in the paragraph regarding different scientific reports about the influence of fucoidan Mw on its biological effects. The required clarification was provided through the following statements: (i) the high molecular weight fucoidans have lower therapeutic effects; (ii) both high and low molecular weight fucoidans could have similar biological activity; (iii) some very low molecular fucoidans are non-active.

Point 4. Sections 4 and 5 are lacking of important details. Authors should discuss the relation of fucoidan structure (Mw, sulfate content, monosacharides, etc) on activity. The phrases like "Fucoidan from Fucus vesiculosus also diminished the secretion of TNF-α and IL-1β in these cells and inhibited the neutrophil infiltration..."; "Methacrylated fucoidan diminished the NO release and CD86 expression. CD86 are co-stimulators of the interaction between antigen-presenting cells and T cells and increased level of CD86 leads to enhanced immune response..."; "Fucoidan also attenuated the increased synthesis of PGE2 and pro-inflammatory mediators such as TNF-α, IL-1β, and IL-6..." etc are not informative. What was a structure of fucoidans mentioned, at which doses activity was observed, what was IC50??? Al these details should be provided. For improving of this part I suggest to authors to prepare a tables for each sections 4.1.1-4.1.5 in which indicate source of fucoidan; its composition; concentration tested, IC50 or concentration range in which activity was observed; type of activity (inhibition, stumulation, etc).

Reply to point 4: We added the suggested information about the concentration in Table 1. However, we failed to obtain enough information about IC50 and this parameter was not included in the table. IC50 data are included in the text if they are available. Regarding the structure of fucoidan, the information was also scarce. Nevertheless, we included a column (title: Compound), which describes the type of fucoidan.

Point 5. For sections 4.2.1-4.2.8 I would suggest to summarise data in tables in which indicate fucoidan source, fucoudan composition, doses tested, activity observed in numbers. For example "Fucoidan treatment significantly decreased the blood serum levels of TNF-α and IL-6 in rats with a model of myocardial infarction" - how model of miocardial infarction was formed? which doses of fucoidan were tested? how fucoidan was administered to animals, was it single or multiple doses administration? for how many % "decreased the blood serum levels", are there positive control? what was efficacy of control comparing with fucoidan? All these data should be included in the tables.

Reply to point 5: The information was included in Table 2 except the description of the myocardial infarction model. This description was added to the text (lines 365-366). Due to the multiple doses tested and the wide range of cytokines included in each experiment, the authors refrained from providing particular numbers of the activity of each dose of fucoidan. However, we included a separate column with the effective doses in each experiment. Further details about the statistical differences between control and tested groups could be found in the respective references.

Point 6. The section 4.3 should be extended with data about type of clinical trial (randomised, placebo controlled, open/blind, number of patients, distribution (male/famale, age), doses, treatment schedule, etc. The resulkts should be provided in numbers as well. How many patients responded on therapy, "reduced the pain, stiffness, difficulty with physical activity and overall symptom severity..." - please provide real data in numbers.

Reply to point 6: We made the requested changes and added the necessary information.

Point 7. Similar data are required for section 5.

Reply to point 7: The suggested details are included in tables 3 and 4 (in the current Section 7).

Point 8. The conclusion should be rewrited based on results presented in suggested Tables.

Reply to point 8: The conclusion is changed based on the presented results in the Tables.

Point 9. The section Material and methods is lacking. How authors have collected information? Which sources were used, which time frame was used for literature.

Reply to point 9: The section “Materials and methods” is added.

Point 10. I would suggest to include in the review and discuss results described in the next recently published papers: https://doi.org/10.1016/j.ijbiomac.2020.04.012; https://doi.org/10.3389/fmars.2016.00129; https://doi.org/10.3390/md18050275; https://doi.org/10.4162/nrp.2017.11.1.3

Reply to point 10: We are grateful for the suggestions. The article https://doi.org/10.1016/j.ijbiomac.2020.04.012 was already included and discussed (under previous reference number 17, current number 16). The other suggested articles are included in this revision as numbers 29, 35, and 54.

Point 11. I would suggest to include special sub-section about pharmacokinetic of fucoidan. Pharmacokinetic help with interpretation of results observed. The bioavailability is essential for understanding of any pharmacological activity.

Reply to point 11: The authors would like to thank the reviewer for the suggestion. We added the pharmacokinetic of fucoidan as Section 5.

Reviewer 2 Report

This review paper reports some interesting information on Immunomodulatory and anti-inflammatory effects of fucoidan. The review paper is well structured and presented some original information. Please consider the following comments to revise the paper:

  • In his paper the authors described a lot of unnecessary details regarding the immune system and the Inflammation reaction. This detailed information may be reduced
  • Figure1: Please reupload a high-quality figure
  • It is interesting to present a figure with Putative mechanisms of anti-inflammatory activity of fucoidan. Is it possible to add a similar figure with fucoidan target diseases?
  • Please delete “. Int. » in reference 59 (the journal abbreviation)

Author Response

Response to Reviewer 2:

The authors would like to thank the Reviewer 2 for the proposed changes in the manuscript. It was really helpful for us and all suggested corrections allowed us to improve the quality of this manuscript.

Please find below our answers to the comments of the Reviewer 2. All corrections, made in the revised manuscript, are marked in blue color.

Point 1. In his paper the authors described a lot of unnecessary details regarding the immune system and the Inflammation reaction. This detailed information may be reduced.

Reply to point 1: We made the suggested changes.

Point 2. Figure1: Please reupload a high-quality figure.

Reply to point 2: A high-quality figure is re-uploaded (current Figure 2).

Point 3. It is interesting to present a figure with Putative mechanisms of anti-inflammatory activity of fucoidan. Is it possible to add a similar figure with fucoidan target diseases?

Reply to point 3: Thank you for the suggestion. A figure regarding fucoidan target diseases is added (current Figure 1).

Point 4. Please delete “. Int. » in reference 59 (the journal abbreviation).

Reply to point 4: The abbreviation was corrected. Thank you for pointing it out.

Round 2

Reviewer 1 Report

Authors have responded most of my questions and have significantly revised and updated the manuscript.

I have one suggestion: please update section 5 with next reference and discuss it:  https://doi.org/10.3390/md17120687

Author Response

The authors would like to thank the Reviewer 1 for his notes. The suggested publication is discussed in section 5 and the discussions made are marked in blue color in the manuscript.
